# Eliminating Nonuniform Smearing and Suppressing the Gibbs Effect on Reconstructed Images †

**Valery Sizikov \*, Aleksandra Dovgan and Aleksei Lavrov**

Faculty of Software Engineering and Computer Technigue, ITMO University, Saint-Petersburg 197101, Russia; aleksandra-dv@yandex.ru (A.D.); lavrov@itmo.ru (A.L.)

\* Correspondence: sizikov2000@mail.ru

† This paper is an extended version of our report: Sizikov, V.; Dovgan, A.; Lavrov, A. "Suppressing the Gibbs Effect on Reconstructed Images" in the Majorov International Conference on Software Engineering and Computer Systems (MICSECS 2019), Saint-Petersburg, Russia, 12–13 December 2019.

**Abstract:** In this work, the problem of eliminating a nonuniform rectilinear smearing of an image is considered, using a mathematical- and computer-based approach. An example of such a problem is a picture of several cars, moving with different speeds, taken with a fixed camera. The problem is described by a set of one-dimensional Fredholm integral equations (IEs) of the first kind of convolution type, with a one-dimensional point spread function (PSF) when uniform smearing, and by a set of new one-dimensional IEs of a general type (i.e., not the convolution type), with a two-dimensional PSF when nonuniform smearing. The problem is also described by a two-dimensional IE of the convolution type with a two-dimensional PSF when uniform smearing, and by a new two-dimensional IE of a general type with a four-dimensional PSF when nonuniform smearing. The problem of solving the Fredholm IE of the first kind is ill-posed (i.e., unstable). Therefore, IEs of the convolution type are solved by the Fourier transform (FT) method and Tikhonov's regularization (TR), and IEs of the general type are solved by the quadrature/cubature and TR methods. Moreover, the magnitude of the image smear, $\Delta$, is determined by the original "spectral method", which increases the accuracy of image restoration. It is shown that the use of a set of one-dimensional IEs is preferable to one two-dimensional IE in the case of nonuniform smearing. In the inverse problem (i.e., image restoration), the Gibbs effect (the effect of false waves) in the image may occur. This may be an edge or an inner effect. The edge effect is well suppressed by the proposed technique, namely, "diffusing the edges". The inner effect is difficult to eliminate, but the image smearing itself plays the role of diffusion and suppresses the inner Gibbs effect to a large extent. It is shown (in the presence of impulse noise in an image) that the well-known Tukey median filter can distort the image itself, and the Gonzalez adaptive filter also distorts the image (but to a lesser extent). We propose a modified adaptive filter. A software package was developed in MATLAB and illustrative calculations are performed.

**Keywords:** smeared image; nonuniform rectilinear smear; integral equations; spectral method; edge and inner Gibbs effects; MATLAB

## 1. Introduction

The elimination of image smearing, a problem of distorted image processing, is commonly undertaken by mathematical- and computer-based approaches ([1–7], et al.).

Smearing refers to image distortion caused by the shift of the image-recording device (i.e., photo camera, video camera, telescope, microscope, etc.), the mismatch of the movement of the observation object (i.e., the Earth's surface, details on a conveyor belt), and the tracking device or the movement of the object itself (e.g., a patient in a tomograph, or an animal, car, or plane) for exposure time [4,7].

The problem of smear elimination via a mathematical method consists of two tasks: a direct problem (computer smear modeling) and an inverse problem (smear eliminating by equation solving) [7] (pp. 106–109). However, the inverse problem is very sensitive to the accuracy of the knowledge of smear magnitude, Δ (as shown, for example, in [8]), especially if the smear is variable, i.e., Δ = Δ(x), where x is the coordinate along the smear [5,6]. In this paper, we develop the "spectral method" proposed in [7–9] and use it to determine the smear Δ and Δ(x).

In this paper, we focus on a rarely considered form of image smearing, namely, rectilinear, which is nonuniform when the camera or object moves during the exposure, as well as on various types of integral equations used in the inverse problem. We give special attention to the Gibbs effect (the effect of false waves), which often appears on images [4,5,7], and noise filtering.

The authors of several publications [1–4,7–9] et al. consider the variant of uniform rectilinear image smearing, as well as that of arbitrary (nonuniform curvilinear) smearing, using a "blind" deconvolution method [10] (p. 192), [11,12], but less often consider an intermediate variant of nonuniform rectilinear smearing [1,2,5,6] as an actual variant.

The main goal of this work is a comparative consideration of two variants of rectilinear image smearing (uniform and nonuniform) and suppression of the Gibbs effect, which reduces the restored image quality.

*Example*: a smeared image of movable objects (e.g., athletes/runners, animals, cars, or planes) moving with different speeds, taken with a fixed camera. Note that a variant was considered in [1,2] in which a camera (or object) moved during the exposure rectilinearly with a certain speed $v(t)$, where $t$ is time. This variant is quite complicated, since it requires solving the nonlinear equation $\delta(t(x)) = x$ with respect to $t$, where $\delta(t) = \int_0^t v(t\prime)\,dt\prime$ [6]. In this paper and in [5,6], a simpler variant of smear is considered: Δ = Δ(x), where x is the coordinate along the smear.

However, it is not possible from [5,6] to determine the dependence Δ(x) from a smeared image. In this paper, we propose such a method ("divided spectra") based on the spectral method.

The article is organized as follows. Section 2 describes the uniform rectilinear image smearing mathematically as a direct problem (integral calculation) and an inverse problem (image restoration via solving a set of one-dimensional IEs or one two-dimensional IE by the Fourier transform (FT) and Tikhonov's regularization (TR) method). Section 3 describes the elimination of nonuniform rectilinear image smearing via solving a set of one-dimensional IEs or one two-dimensional IE of a general type (not convolution type) by the method of quadrature/cubature and TR. In Section 4, an illustrative numerical example is given: three cars moving with the same smear Δ. The smear is determined by the spectral method. The results of image restoration by the FT and TR methods are presented. The question of restoration error is stated. In Section 5, solution results for examples are presented in the case of various smears of automobiles. A method using divided spectra for determining the dependence Δ(x) is proposed. Section 6 describes the impulse noise filtering and its influence on the image itself. A modified median filter for impulse noise suppression is proposed. In all sections, we use the technique of diffusing the image edges that we proposed for suppressing the Gibbs effect. In Appendix A we also clarify why we used complex term—smear (motion blur).

## 2. Mathematical Description of Uniform Rectilinear Smearing of the Image

Let us recall the well-known case of image uniform smearing [4,7,9] and consider the direct and inverse problems.

### 2.1. Direct Problem

The direct problem of uniform rectilinear smearing of an image is to calculate the integral [6]

$$g_y(x) = \frac{1}{\Delta} \int_x^{x+\Delta} w_y(\xi)\,d\xi, \tag{1}$$

where $\Delta$ = const is the smear magnitude; the axes $x$ and $\xi$ are directed along the smear, and the axis $y$ is perpendicular to the smear (plays the role of the parameter); $w_y(\xi)$ is the given non-smeared image, and $g_y(x)$ is the calculated (modeled) smeared image in each $y$-line. In order to calculate $g$ according to (1), we developed in MATLAB the main program Autos.m and m-functions smearing.m [7] for the arbitrary smear angle $\theta$ and smear.m [5] for $\theta = 0$. In addition, the MATLAB system has m-functions fspecial.m and imfilter.m for modeling $g$ [10].

It should be noticed that the description of the smearing problem by an integral (and, as a consequence, by an integral equation, see below) is the most adequate mathematical description of the image smearing process.

The following Algorithm 1 in the form of pseudo code describes solving the direct problem according to [7] (pp. 116–118).

---

**Algorithm 1.** The direct problem of uniform rectilinear smearing of an image.

---

Input: exact (undistorted) image $w$

(1)     Assignment of smear magnitude $\Delta$ and smear angle $\theta$.
(2)     Turn of the image $w$ about angle $\theta$ and introduction of boundary conditions (BCs), truncation or diffusion of edges
(3)     Calculation of the smeared image $g$ line-by-line in the discrete form: $g_j(i) = \frac{1}{\Delta+1}\sum_{k=i}^{i+\Delta} w_j(k)$,
        $j = 1, 2, \ldots, m, i = 1, 2, \ldots, N$, where $j$ and $i$ are the numbers of rows and columns of the turned image, and $N = n - \Delta$ when truncating and $N = n + \Delta$ when diffusing the image edges.
(4)     Inverse turn of the image $g$.

Output: $g$

---

### 2.2. Inverse Problem

The inverse (more important and complex) problem can be solved by *two approaches*.

In the *first approach*, in order to eliminate the smearing, we solve a set of one-dimensional Fredholm integral equations (IEs) of the first kind of convolution type (for each value of $y$) [6,7,9]:

$$\int_{-\infty}^{\infty} h(x - \xi)\, w_y(\xi)\, d\xi = g_y(x), \quad -\infty < x < \infty. \tag{2}$$

where

$$h(x) = \begin{cases} 1/\Delta, & -\Delta \leq x \leq 0, \\ 0, & \text{otherwise.} \end{cases} \tag{3}$$

The IE (2) is obtained from the relation (1); axes $x$ and $\xi$ are directed along the smear and axis $y$ is perpendicular to the smear; $h$ is the kernel of IE mathematically, and the point spread function (PSF) physically and technically [3,4,7,11,13]. The function $h$, as a rule, is difference, or spatially invariant, which means that the smearing is uniform and the smear magnitude $\Delta$ is the same at all points of the image ($\Delta$ = const).

The problem of solving IE (2) is ill-posed (unstable) [14–17]. We use the stable Tikhonov regularization method (TRM, TR) with the Fourier transform (FT) [4,7,9,16,18]:

$$w_{\alpha y}(\xi) = \frac{1}{2\pi} \int_{-\infty}^{\infty} W_{\alpha\, y}(\omega)\, e^{-i\omega\xi}\, d\omega, \tag{4}$$

where

$$W_{\alpha\, y}(\omega) = \frac{H(-\omega)\, G_y(\omega)}{\left|H(\omega)\right|^2 + \alpha\, \omega^{2p}} \tag{5}$$

is the regularized Fourier spectrum (FS) of the solution; $H(\omega) = F(h(x))$ and $G_y(\omega) = F(g_y(x))$ are Fourier spectra of functions $h(x)$ and $g_y(x)$ ($H(\omega)$ is the transfer function of a system [9]), where $F$ is the

sign of FT; $\alpha > 0$ is the regularization parameter; $p \geq 0$ is the regularization order (usually $p = 1$ or $p = 2$). Several approaches have been developed for choosing the regularization parameter $\alpha$: the discrepancy principle, the method of training examples, the selection method, and others [2,7,14,17,18]. In order to calculate the reconstructed image using expressions (4) and (5), we have developed the m-function desmearingf.m [10]. In addition, the MATLAB system has the m-function deconfreg.m [10].

The following Algorithm 2 describes the solution of inverse problem (the first approach).

---

**Algorithm 2.** The inverse problem of the uniform rectilinear smearing (the first approach)

---

Input: smeared image $g(x, y)$, where $x$ and $y$ are directed horizontally and vertically, respectively.

(1)　Calculating the Fourier spectrum $G(\omega_1, \omega_2) = F(g(x, y))$, where $x$ is horizontally, and $y$ is vertically.
(2)　Determining the smear $\Delta$ and angle $\theta$ values according to the Fourier spectrum (by the spectral method).
(3)　Calculating the PSF $h(x)$ according to (3), where $x$ is directed along the smear and $y$ is perpendicular to the smear.
(4)　Calculating the Fourier spectra $H(\omega) = F(h(x))$ and line-by-line $G_y(\omega) = F(g_y(x))$.
(5)　The choice of the regularization parameter $\alpha$ by some approach.
(6)　Calculating line-by-line (for every $y$) the $W_{\alpha\,y}(\omega)$ – the regularized Fourier spectrum of the desired solution $w_{\alpha\,y}(\xi)$ according to (5).
(7)　Calculating the restored image as IFT line-by-line: $w_{\alpha\,y}(\xi) = F^{-1}\big(W_{\alpha\,y}(\omega)\big)$ according to (4)
(8)　Inverse turn of the image and obtaining $w(x, y)$, where $x$ is horizontally, and $y$ is vertically

Output: $w(x, y)$.

---

In the *second approach*, to eliminate the smearing (as well as defocusing), the two-dimensional Fredholm integral equation of the first kind of convolution type is used (cf. (2)) [2,6,7]:

$$\int_{-\infty}^{\infty} \int_{-\infty}^{\infty} h(x - \xi, y - \eta)\, w(\xi, \eta)\, d\xi\, d\eta = g(x, y), \quad -\infty < x, y < \infty \tag{6}$$

In this equation, the $x$ and $\xi$ axes are directed horizontally, and $y$ and $\eta$ are directed vertically downward. PSF $h$ is displayed on the plane $(x, y)$ as a narrow strip [6,7] (p. 112).

In this approach, the direct problem is calculated using the m-functions fspecial.m and imfilter.m [10]. However, the solution of the two-dimensional IE (6) (the inverse problem) by the TR method and the two-dimensional FT is equal to $w_\alpha(x, y) = F^{-1}(W_\alpha(\omega_1, \omega_2))$, where $F^{-1}$ is the inverse Fourier transform (IFT), or

$$w_\alpha(x, y) = \frac{1}{4\pi^2} \int_{-\infty}^{\infty} \int_{-\infty}^{\infty} W_\alpha(\omega_1, \omega_2)\, e^{-i(\omega_1 x + \omega_2 y)}\, d\omega_1\, d\omega_2, \tag{7}$$

where $W_\alpha(\omega_1, \omega_2)$ is the regularized spectrum (two-dimensional FT) of the solution, equal to

$$W_\alpha(\omega_1, \omega_2) = \frac{H^*(\omega_1, \omega_2)\, G(\omega_1, \omega_2)}{\big|H(\omega_1, \omega_2)\big|^2 + \alpha\, (\omega_1^2 + \omega_2^2)^p} \tag{8}$$

where $H(\omega_1, \omega_2) = F(h(x, y))$, $G(\omega_1, \omega_2) = F(g(x, y))$. The MATLAB contains the m-function deconvreg.m [10] for solving IE (6) by the TR and FT methods according to (7), (8).

The following Algorithm 3 describes the solution of the inverse problem (the second approach).

---

**Algorithm 3.** The inverse problem of the uniform rectilinear smearing (the second approach)

---

Input: smeared image $g(x, y)$, where $x$ and $y$ are directed horizontally and vertically respectively

(1)   Calculating the Fourier spectrum $G(\omega_1, \omega_2) = F(g(x, y))$.
(2)   Determining the smear magnitude $\Delta$ and the smear angle $\theta$ from the spectrum (by the spectral method).
(3)   Building the PSF $h(x, y)$ in the form of a strip with length $\Delta$ at angle $\theta$.
(4)   Calculating the Fourier spectrum $H(\omega_1, \omega_2) = F(h(x, y))$.
(5)   Choice of the regularization parameter $\alpha$ and regularization order $p$ in some approach.
(6)   Calculating $W_\alpha(\omega_1, \omega_2)$ – the regularized Fourier spectrum of the solution according to (8).
(7)   Computation of the restored image in the IFT form: $w_\alpha(x, y) = F^{-1}(W_\alpha(\omega_1, \omega_2))$ according to (7).

Output: $w(x, y)$

---

In order to compare the various approaches below, we give the well-known Formulas (1)–(8).

## 3. Mathematical Description of the Nonuniform Rectilinear Smearing

Taking into account the expressions (1)–(8), we describe the nonuniform rectilinear smearing of image along the smear direction. Let us consider *two approaches*.

### 3.1. The First (Time) Approach [1,2,6]

In this approach, the speed of moving object (or camera) is believed to be known as a function $v(t)$ of time $t \in [0, \tau]$, where $\tau$ is the exposure time. This approach was considered in detail in [6], and we consider the second, simpler approach.

### 3.2. The Second (Spatial) Approach [5,6]

Suppose that we have determined a dependence $\Delta = \Delta(x)$ of the smear $\Delta$ on the $x$ coordinate directed along the smear. This dependence may be determined from the smeared image in some approach, for example, by the method of "blind" deconvolution [11] or by the spectral method [9] (see Section 4).

#### 3.2.1. The Direct Problem

In this case, the PSF $h$ is not difference, or spatially invariant, and *the direct problem* is written in the form (cf. (1)):

$$g_y(x) = \frac{1}{\Delta(x)} \int_x^{x+\Delta(x)} w_y(\xi) \, d\xi \tag{9}$$

To calculate $g_y(x)$ according to (9), we have developed the m-function smear_n.m [5].

#### 3.2.2. The Inverse Problem

The *inverse problem* in the frames of the second approach is written as a set of one-dimensional Fredholm integral equations of the first kind of general type (not in the convolution type) for each value of $y$ [5,6]:

$$Aw_y \equiv \int_a^b h(x, \xi) \, w_y(\xi) \, d\xi = g_y(x), \quad c \leq x \leq d \tag{10}$$

where $A$ is integral operator; $[a, b]$ and $[c, d]$ are limits for $\xi$ and $x$. The PSF $h$ is written as

$$h(x, \xi) = \begin{cases} 1/\Delta(x), & x \leq \xi \leq x + \Delta(x), \\ 0, & \text{otherwise.} \end{cases} \tag{11}$$

The FT method cannot be used for solving IE (10), since the IE (10) is not an equation of convolution type. However, the *quadrature method* can be applied, which reduces the IE (10) to some system of linear algebraic equations (SLAE) for each $y$ [7] (p. 126):

$$Aw_y = g_y \tag{12}$$

where $A$ is the matrix, associated with $h$ (the same for all $y$-rows), $w_y$ is the desired vector, $g_y$ is the right-hand side of SLAE. A stable solution of SLAE (12) is provided by the Tikhonov regularization method (TRM) [7]:

$$(\alpha I + A^T A)\, w_{y\,\alpha} = A^T g_y \tag{13}$$

where $\alpha > 0$ is the regularization parameter (the $\alpha$ choice method see in Figure 5 further), $I$ is the identity matrix, $A^T$ is the transposed matrix, and $w_{y\alpha}$ is the regularized solution in each $y$-row, equal to

$$w_{y\,\alpha} = (\alpha I + A^T A)^{-1} A^T g_y \tag{14}$$

In order to realize the formulas (10)–(14) on the PC, we have developed the m-function desmearq_n.m. Note that the quadrature method with Tikhonov's regularization (TR) according to (13), (14) can be also used for solving IE of convolution type (2) with the PSF (3), i.e., for uniform smearing. For that, we have developed the m-function desmearq.m.

The *inverse problem* in the framework of the *second approach* can be also presented in the form of a two-dimensional Fredholm IE of the first kind of general type [18] (cf. (6)):

$$Aw \equiv \int_a^b \int_c^d h(x, \xi, y, \eta)\, w(\xi, \eta)\, d\xi\, d\eta = g(x, y), \; a \le x \le b, \; c \le y \le d \tag{15}$$

Integral Equation (15) can be solved by the *quadrature method* (more precisely, *cubature method*) (cf. [18] p. 167).

According to this method, each of the integrals in (15) is replaced by a finite sum on discrete grids of nodes in $x$, $\xi$, $y$, $\eta$ and we obtain SLAE with a four-dimensional matrix $A$, a two-dimensional right-hand side $g$ and a two-dimensional desired function $w$. To solve such SLAE by the known methods (Gauss, etc.), it is necessary to transform the four-dimensional matrix $A$ into a two-dimensional one, to transform the two-dimensional right-hand side $g$ into a one-dimensional one, to solve such SLAE, and then to transform the resulting one-dimensional solution $w$ into a two-dimensional one.

The solution of a two-dimensional IE by the cubature method took place, for example, in [18] (p. 168), however, with the small matrix $11 \times 11 \times 7 \times 7$, i.e., $11^2 \times 7^2$. If an image $g$ has real dimensions, e.g., $400 \times 400$ pixels, then it is necessary to solve SLAE with the two-dimensional matrix of size $400^2 \times 400^2$, i.e., the matrix of gigantic size 160,000 $\times$ 160,000, which is difficulty implemented even on powerful computers. We see that the cubature method for solving IE (15) is a cumbersome method and its application to restore a nonuniform smeared image is complex.

It is possible to use the iteration methods of Jacobi, Landweber, Friedman, Bakushinsky, et al. [2,15,18] for solving two-dimensional IE (15). These methods usually do not use giant arrays; however, they require successful initial approximation for $w$, the number of iterations (playing the role of a regularization parameter), and other parameters.

As a result, we conclude that the most effective technique in the case of nonuniform smearing is the technique (10)–(14), based on line-by-line image processing by solving one-dimensional IE (10) and SLAE (12) with two-dimensional matrix in each $y$-line and on the subsequent combination of row-wise solutions into a single two-dimensional image.

The following Algorithm 4 describes the line-by-line solution of inverse problem.

---

**Algorithm 4.** The inverse problem for illuminating the row-wise nonuniform rectilinear smear

---

Input: image $g(x, y)$ smeared nonuniformly along $x$
(1) Presentation of a two-dimensional image $g(x, y)$ as a set of one-dimensional images $g_y(x)$.
(2) Determining the nonuniform smear $\Delta(x)$ from the spectrum (by the spectral method).
(3) Calculating a PSF $h(x, \xi)$ according to (11).
(4) Writing matrix $A$ and vectors $g_y$ of a SLAE.
(5) Choosing the regularization parameter $\alpha$ in some approach.
(6) Solving the regularized SLAE (13) in each $y$-row.
(7) Obtaining the regularized solution $w_{y\alpha}$ according to (14).
(8) Forming image $w_\alpha(x, y)$ from a set of row-wise solutions $w_{y\alpha}$.
Output: $w_\alpha(x, y)$.

---

## 4. Illustrative Example

The following *numerical example* is solved. Figure 1 shows semi-model grayscale (gray) initial image J of three cars (file Autos.png).

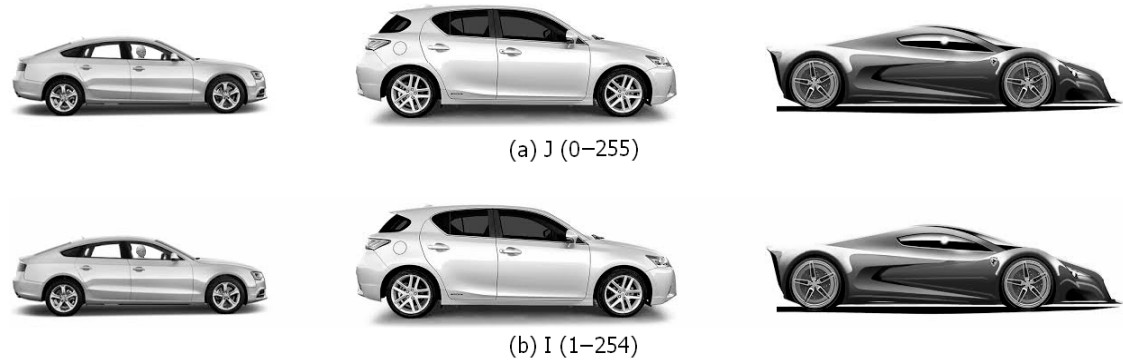

(a) J (0−255)

(b) I (1−254)

**Figure 1.** Initial (undistorted) image of stationary cars (143 × 1307 pixels). (**a**) Image J with intensities from 0 to 255; (**b**) image I with intensities from 1 to 254.

Image J has intensities from 0 to 255, which causes inconvenience in filtering impulse noise, which has values 0 and 255 (see details in Section 6).

To avoid this inconvenience, we replace the intensities 0 and 255 of the image J with the values 1 and 254. We obtain the image I with intensities from 1 to 254.

We use the image I 143 × 1307 pixels for further processing. Note that the driver's head is visible in the left car, which (along with the wheels) will be one of the reference points in image processing.

### 4.1. Uniform Smearing of Image Using Boundary Conditions

Further, we assume that the cars move at the same speed and therefore give the same smear in the image: $\Delta$ = const = 20 pixels or the cars are stationary, but the camera moves uniformly during the exposure. Image I is smeared horizontally according to (1) using the head program Autos.m as well as the m-functions fspecial.m and imfilter.m at $\Delta$ = 20. In this case, a "boundary condition" is used in the m-function imfilter.m with the option 'circular'. Figure 2a presents a smeared image (see smeared driver's head and wheels).

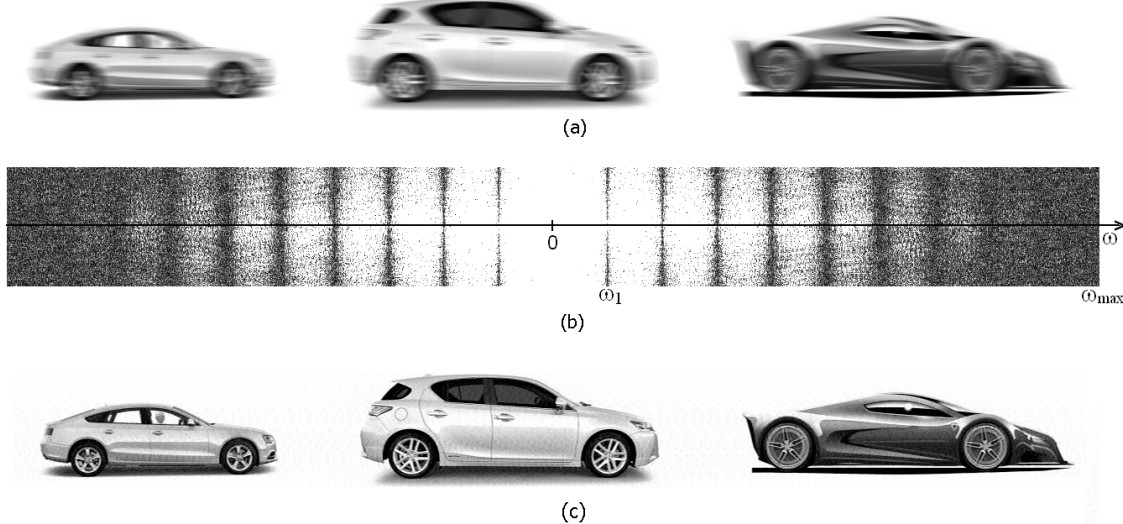

**Figure 2.** Image $143 \times 1307$ smeared uniformly (**a**); two-dimensional centered FT (spectrum) in modulus of image $143 \times 1307$ (**b**); image $143 \times 1307$ restored by the Fourier transform (FT) and Tikhonov regularization (TR) method (**c**).

In order to restore the image by solving inverse problem, it is necessary to know the angle $\theta$ and the smear $\Delta$. It is possible to estimate these parameters visually, but only with an error, or it is impossible to estimate them at all. As we showed in [8,19], even the error of $\theta$ in 1–2 degree and the error of $\Delta$ in 1–2 pixel lead to the significant error (~10%) in image restoration even by Tikhonov's regularization or Wiener's parametric filtering method.

In [7–9,19], we have developed the so-called "spectral method", which allows us to determine the distortion type (smearing, defocusing, etc.) and estimate the distortion parameters based on the Fourier transform (Fourier spectrum) of the distorted image. Let us use the spectral method. Figure 2b presents the Fourier spectrum in modulus $|G(\omega)|$ of distorted image showed in Figure 2a. As follows from Figure 2b, the image in Figure 2a is uniformly smeared (this follows from a theory and a large number of Fourier spectra of smeared, defocused, and other images [3,6–9,19–21]). In addition, one can estimate the smear magnitude $\Delta$ by the formula

$$\Delta = 2\frac{\omega_{max}}{\omega_1} \tag{16}$$

where $\omega_1$ and $\omega_{max}$ are the first and last zeros of the transfer function $H(\omega) = F(h(x))$ (see the details in [9]). For several measures in Figure 2b, we get on average $\Delta = 20.08 \pm 0.05$ pixels, which is close to the exact value of $\Delta = 20$. Using the value of $\Delta$, we restored the image of cars (Figure 2c) by the FT and TR method according to (6)–(8) using m-function deconvreg.m with $\alpha = 10^{-4.2}$ ($\alpha$ value is chosen by the minimum of curve 1 in Figure 5); relative error is $\sigma_{rel} = 0.0359$ (according to (17)).

*4.2. Uniform Image Smearing with Truncation*

The Boundary Conditions (BCs) [10,22] used in m-function imfilter.m are an artificial technique intended for determining the intensities of an undistorted image outside its edges. Instead of BCs, we have proposed [7,23] the image truncation technique, which does not require extra boundary data. Figure 3a presents the image $143 \times 1267$ which is smeared uniformly with a truncation modeled using m-function smear.m (with the option 'truncation'). Figure 3b presents the image restored by the FT and TR method using m-function desmearq.m with $\alpha = 10^{-1.2}$ and $\sigma_{rel} = 0.269$. We see the significant edge and smaller inner Gibbs effects in Figure 3b. This is due to the sharp jump of intensity on the edges of image if it is truncated. Nevertheless, the truncation technique can be effective in the case of a finite image (a space object on the dark background [7] (p. 105), [22], etc.).

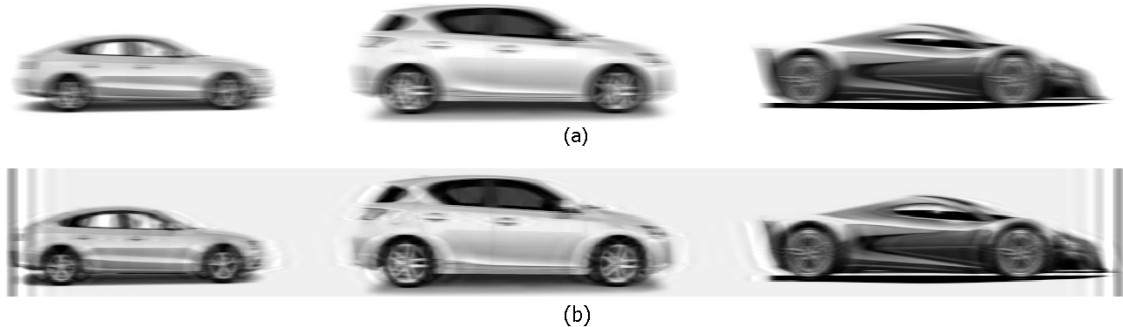

(a)

(b)

**Figure 3.** Image smeared uniformly with truncation (**a**); restored image (**b**).

### 4.3. Uniform Image Smearing with Diffusing the Image Edges

In [7,23], we have proposed the approach of diffusing the smeared image edges to suppress the Gibbs edge effect. The Gibbs effect (distortion of the "ringing" type) is due to a stepwise change in intensity of the Heaviside step function type. Mathematically, the Gibbs effect can be explained by the fact that the FT of the step function gives the sinc function with side fluctuations. In order to suppress these false fluctuations, in [10] (p. 185), it was proposed to use the m-function edgetaper.m, which diffuses the image edges. However, modeling has shown that this function does not sufficiently diffuse the edges, and the foregoing Algorithm 1 is more effective. At the same time, the Gibbs inner effect is eliminated more difficult, but the image smearing itself reduces the intensity jump and reduces the Gibbs inner effect (see middle of Figure 3b).

The technique of diffusing the image edges resembles smoothing the edges of the transfer function $H(\omega)$, as well as the image smoothing by Butterworth low-frequency filter [4] (p. 265–267), [10] (p. 143), etc. However, these filters enhance smoothing the edges in the frequency domain, and our technique realizes diffusing the edges of the image itself immediately, which leads to a greater suppression of the Gibbs effect.

The diffusing of edges leads to the smooth decrease of intensity on the smeared image edges and to suppressing the Gibbs effect on the restored image. Figure 4a presents the uniformly smeared image with the addition of diffusing its edges via m-function smear.m (with the option 'diffusion'). Figure 4b presents the result of image restoration of cars by the FT and TR method according to (12)–(14) with the m-function desmearq.m. We see that the introduction of diffusing the edges leads to the image restoration without the Gibbs effect and to almost exact restoration.

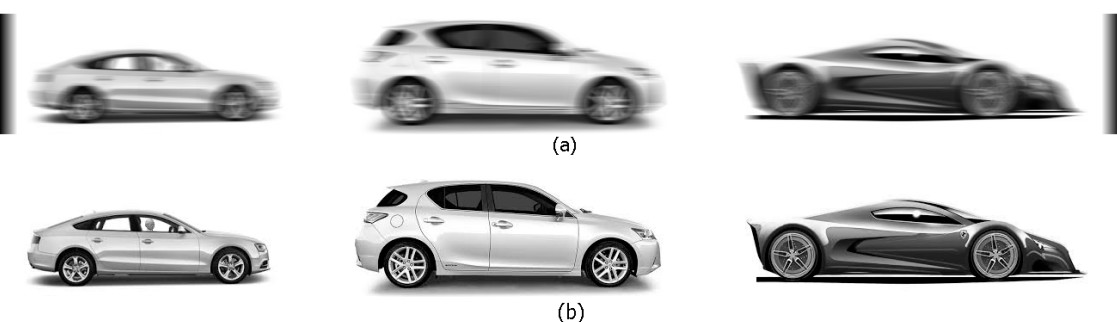

(a)

(b)

**Figure 4.** Image smeared uniformly with diffusing the edges (option 'diffusion'). (**a**)—smeared image $143 \times 1327$ ($\Delta = 20$ pixels); (**b**)—restored image $143 \times 1307$ by TR method ($\alpha = 10^{-8}$), relative error of restoration $\sigma_{rel} \approx 0$.

### 4.4. Error Estimation of Image Restoration

In order to estimate numerically the restoration quality and choose the regularization parameter $\alpha$, we propose a formula for calculating relative error in the form of the standard deviation of the calculated image $\widetilde{w}_\alpha$ from the exact image $\overline{w}$ [9]:

$$\sigma_{\text{rel}}(\alpha) = \frac{\left\|\widetilde{w}_\alpha - \overline{w}\right\|_{L_2}}{\left\|\overline{w}\right\|_{L_2}} = \sqrt{\sum_{j=1}^{M}\sum_{i=1}^{N}\left(\widetilde{w}_{\alpha ji} - \overline{w}_{ji}\right)^2} \Big/ \sqrt{\sum_{j=1}^{M}\sum_{i=1}^{N}\overline{w}_{ji}^2} \tag{17}$$

where $M$ is the number of rows and $N$ is the number of columns in the image. Such an expression for the image error can be used only in the case of model image processing when $\overline{w}$ is known (e.g., the image in Figure 1b). Figure 5 presents the curves $\sigma_{\text{rel}}(\alpha)$ for different variants of solutions. One can choose the value of $\alpha$ from the minima of these curves.

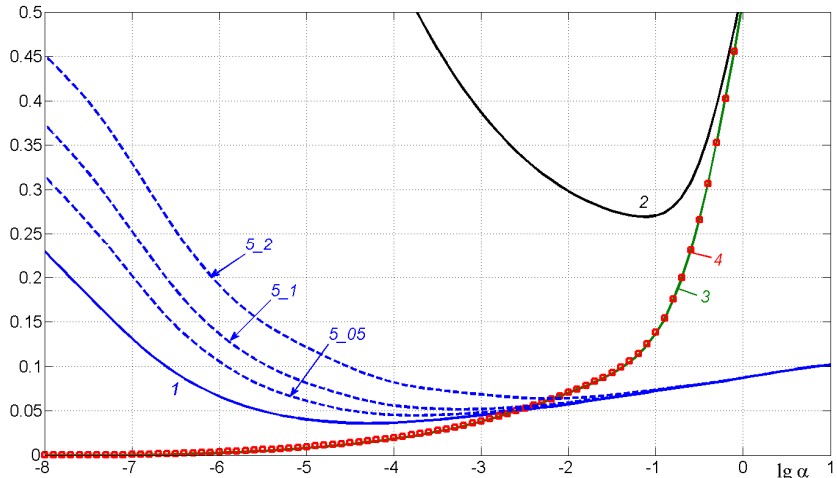

**Figure 5.** The relative error of image restoration $\sigma_{\text{rel}}(\alpha)$. *1*—FT and TR method (see Figure 2c); *2*—FT and TR method with truncation (Figure 3b); *3*—FT and TR method with diffused edges (Figure 4b); *4*—quadrature and TR method with diffusing and division (see Figure 9); *5* – FT and TR method with noise, d = 0.005, 0.01, 0.02 (Figure 10e).

One can also use the following expression for restoration error:

$$\text{PSNR} = 10\lg\left(\overline{w}_{\max}MN / \|w - \overline{w}\|_{L_2}^2\right) \tag{18}$$

where PSNR is the so-called peak signal-to-noise ratio, used widely in engineering, acoustics, and other fields. However, formula (17) is more convenient and obvious for image processing. Indeed, if, for example, the error according to (17) equals $\sigma_{\text{rel}} = 0.035$, then this means that the restoration error is 3.5%, and this value does not depend on the system of units $w$, which is convenient and clear. Although, if we use the formula (18) and get an error, for example, 8 dB, it becomes difficult to judge whether this is a large or small error in image restoration.

## 5. Direct and Inverse Problems of Nonuniform Image Smearing

The next step is the nonuniform image smearing. We believe that the cars move with different speeds and therefore they have different smears on the image, for example, $\Delta = \Delta_1 = 15$ pixels for the left car, $\Delta = \Delta_2 = 20$ pixels for the middle car and $\Delta = \Delta_3 = 25$ pixels for the right car.

As a result, smear $\Delta(x)$ is a piecewise constant function:

$$\text{if } (i <= 360)\ \Delta = 15;\ \text{elseif } (\ i <= 820)\ \Delta = 20;\ \text{else } \Delta = 25; \tag{19}$$

where $i = 1 \ldots 1307$ is the number of discrete reading along $x$.

Figure 6 shows an image smeared nonuniformly according to (9) and (19) with diffusing the edges for suppressing the Gibbs edge effect using m-function smear_n.

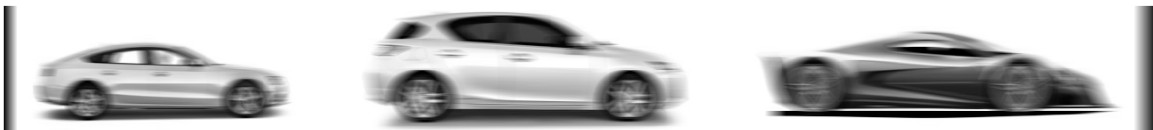

**Figure 6.** Image smeared nonuniformly with diffusing the edges.

In order to solve the inverse problem of image restoration, one needs to know the dependence (19) $\Delta(x)$ of the smear from $x$. However, we usually do not know this dependence in practice, or we can visually estimate it according to Figure 6 only approximately. To determine $\Delta(x)$, we use the spectral method, as in Figure 2b. Figure 7 presents the Fourier spectrum in modulus $|G(\omega)|$ of the distorted image $g(x, y)$ in Figure 6.

However, Figure 7 shows that the spectrum is the sum of three spectra and this creates difficulties with the definition of $\Delta_1$, $\Delta_2$ and $\Delta_3$. We propose the approach that we call the "approach of divided spectra".

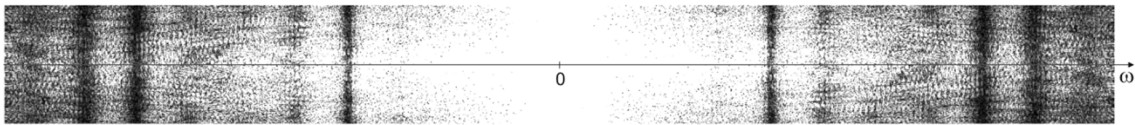

**Figure 7.** FT (Fourier spectrum) in modulus of nonuniformly smeared image presented in Figure 6.

*The Approach of Divided Spectra*

Let us divide the original piecewise-uniform smeared image in Figure 6 into three parts, for example, g1(:,22:344); g2(:,400:800), g3(:,854:1293). To perform such a division is not difficult. Figure 8a–c present the divided parts g1, g2, g3 of the image shown in Figure 6.

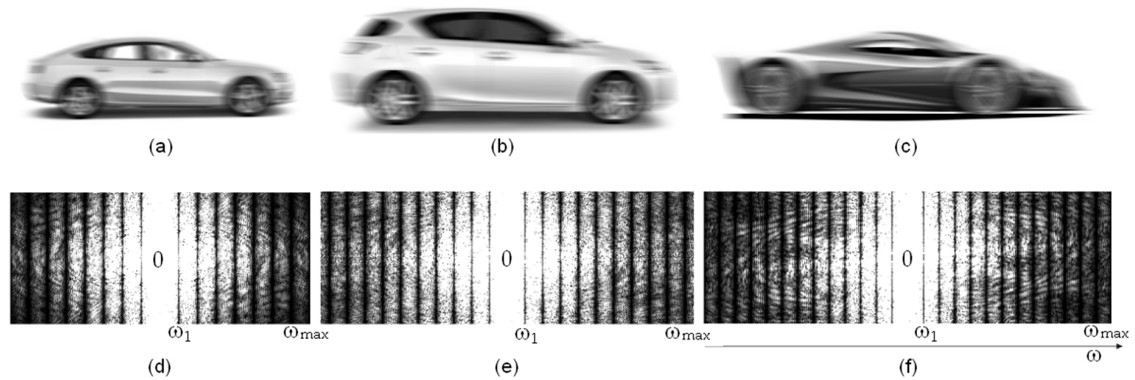

**Figure 8.** Divided parts of image (**a**–**c**); divided spectra in modulus (**d**–**f**).

Next, we calculate the FT (spectrum) in modulus $|G|$ for each part of the image (Figure 8d–f, cf. Figure 2b). From the spectra, we determine the magnitudes of smears by the Formula (16) and obtain (for several measures): $\Delta_1 = 14.95 \pm 0.40$, $\Delta_2 = 20.42 \pm 0.50$, $\Delta_3 = 24.99 \pm 0.60$ or rounded to integer values: $\Delta_1 = 15$, $\Delta_2 = 20$, $\Delta_3 = 25$.

As a result, having determined rather accurately the divided smears $\Delta_1$, $\Delta_2$, $\Delta_3$, and hence the dependence $\Delta(x)$, we solve the inverse problem of a single (without division) image restoration according to (10)–(14), namely, by the quadrature method with Tikhonov's regularization. Figure 9

presents the result of image restoration using the developed m-function desmearq_n.m. The image is restored almost accurately and without the Gibbs effect.

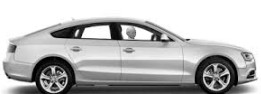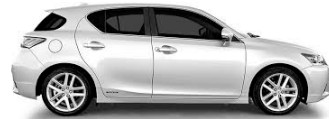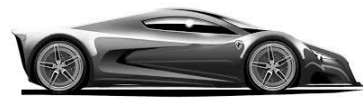

**Figure 9.** Restored single image 143 × 1307, $\sigma_{rel} \approx 0$ (see curve 4 in Figure 5).

## 6. Noise Accounting

In the above results, we do not take into account the influence of noise; therefore, we obtain clear restored images in Figures 2, 4 and 9. Let us consider the influence of noise on the results of image processing for example of impulse noise (of type 'salt & pepper') with intensities 0 and 255 [10,24–26].

In the Tukey median filter [4,7,24,25] (realized in the m-function medfilt2.m), a small window slides over the noisy image and at each position of the window, the intensity of center point of the window $I_0$ is replaced by the median intensity $I_m$. As a result, noise with intensities $I_n = 0$ and 255 is eliminated, but the image itself may be distorted, since, firstly, $I_m$ can be different from $I_0$ and, secondly, $I_0$ can be equal to 0 or 255.

In [4] (p. 360) and [10] (p. 178), an adaptive median filter (the Gonzalez filter) was proposed (m-function adpmedian.m), according to which $I_0$ is replaced by $I_m$ only when $I_0 = 0$ or 255, i.e., the central point is impulse. However, the Gonzalez filter can also distort the image itself when $I_0 = 0$ or 255, although the center point is not an impulse, but the point of the image itself.

We propose a filter that we call a modified median filter (m-function modmed.m). According to this filter, the original noise-free image is supposed to have no points with intensity 0 or 255, i.e., $0 < I_0 < 255$, and replacing $I_0$ with $I_m$ in a noisy image is performed only when $I_0 = 0$ or 255, i.e., when the center point is really an impulse, as in the Gonzalez filter. If the noise-free image haspoints with intensities 0 or 255, then we propose to replace $I_0 = 0$ or 255 with $I_0 = 1$ or 254 in the noise-free image (as in Figure 1).

The following Algorithm 5 describes the operations of the modified median filter.

---

**Algorithm 5.** Noise eliminating by the modified median filter.

---

Input: *g* – image noisy by bipolar impulse noise

(1)　Initial (noise -free) image w of size M×N

(2)　Correction of w: for j = 1:M for i = 1:N if w(j,i) = 0 then w(j,i) = 1; if w(j,i) = 255 then w(j,i) = 254;

(3)　Noising image w by impulse noise and obtaining image g

(4)　Presentation of sliding window m×n (default 3×3)

(5)　The Tukey filter (median intensity at each point): fmed = medfilt2(g,[m n]);

(6)　Initial approximation of filtered image: f = g;

(7)　Noise eliminating: for j = 1:M for i = 1:N if g(j,i) = 0 or g(j,i) = 255 then f(j,i) = fmed(j,i); (replacement only if the point (j,i) is an impulse)

Output: f – image filtered by the modified median filter

---

Figure 10a presents the image taken from Figure 2a and uniformly smeared (Δ = const = 20 pixels). Figure 10b presents an image that is smeared and slightly noisy by impulse noise (the share of noising is d = 0.005, i.e., 0.5%). Figure 10c shows the image cleared from noise by the modified median filter (m-function modmed.m). The elimination of noise seemed to be successful, but the FT (spectrum) in modulus in Figure 10d of image from Figure 10c is insufficiently clear, unlike the clear spectrum in Figure 2b. Consequently, eliminating the image smear in Figure 10c by the FT and TR method using the

m-function deconvreg.m gives an insufficiently accurate result (Figure 10e, $\alpha = 10^{-3.7}$, $\sigma_{rel} = 0.0448$, see curve 5_05 in Figure 5), besides with the Gibbs weak effect.

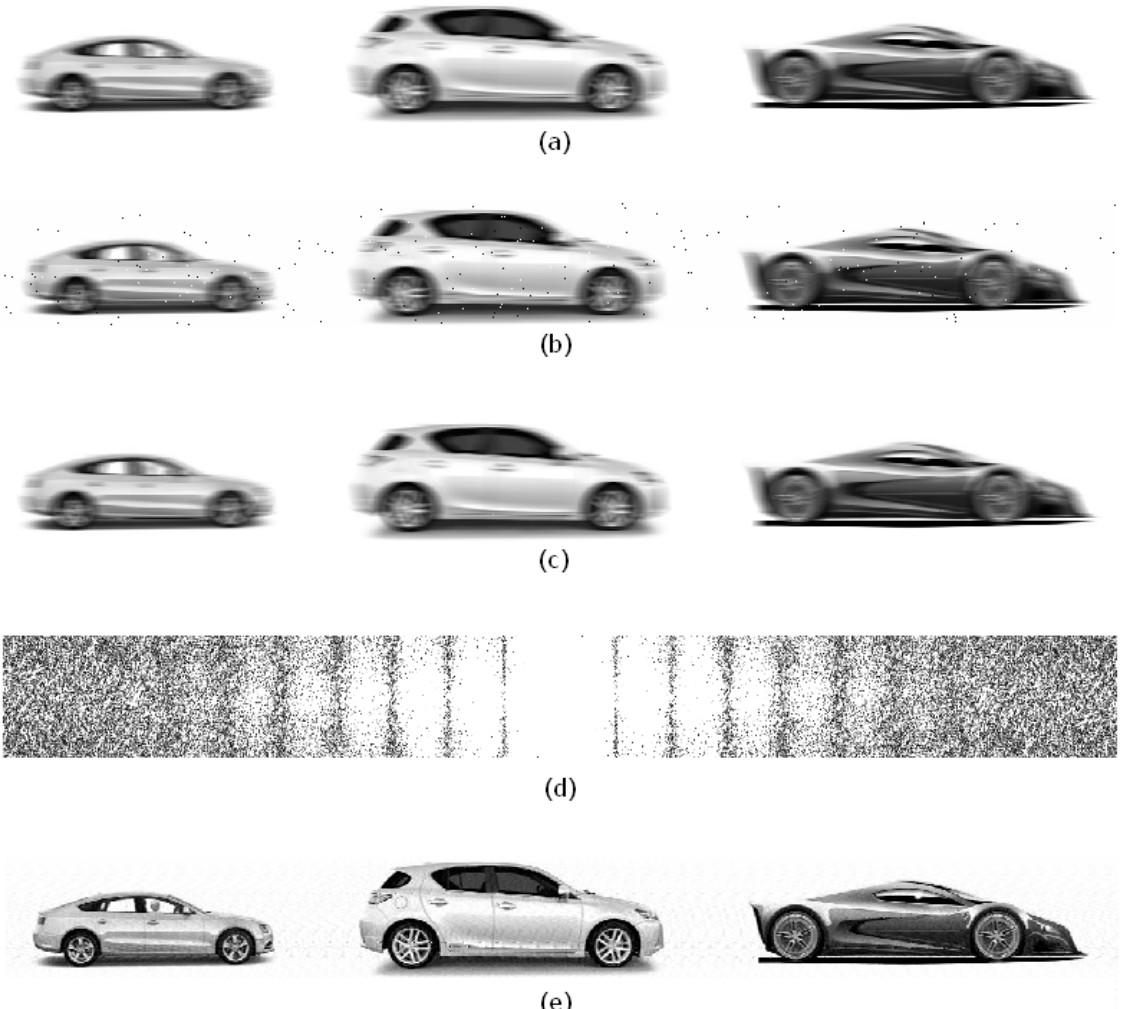

**Figure 10.** Noise influence accounting. (**a**) Image smeared uniformly; (**b**) image smeared uniformly and noisy (d = 0.005); (**c**) image filtered from noise using modmed.m; (**d**) FT (spectrum) in modulus; (**e**) image restored by the FT and TR method.

Note that the intensities in the image in Figure 10a are artificially put equal from 1 to 254, as the modified median filter requires.

We also note that the impulse noise level (the share of noising) is put equal to d = 0.005, 0.01 and 0.02 (see Figures 5 and 10) in order to determine the error dependence of image filtering by the modified median filter from the noise level. We see from a comparison of the images in Figure 10a,c that this filter eliminates noise effectively, but slightly distorts the image itself (Figure 10c). This leads to the noticeable distortion of the spectrum (Figure 10d) and the restored image (Figure 10e).

This means that the Tukey median filter has an error in restoring the center point of a sliding window. If we pass through medfilt2.m even a noise-free image, for example, the smeared image S1 in Figure 10a, we get the image Sf1 = medfilt2(S1, [3 3], 'symmetric'). The image Sf1 is different from S1 on 2.4%, i.e., the Tukey median filter distorts an image even in the noise absence. Since both the Gonzalez filter and our modified filter refer to the Tukey filter, they also distort an image (cf. Figure 10a,c). The relative error of the Tukey filter is $\sigma_{rel} = 0.0244$, the Gonzalez filter is $\sigma_{rel} = 0.0135$, and the modified filter is $\sigma_{rel} = 0.0017$ at noise d = 0.005.

## 7. Conclusions

In this paper, we describe a technique of restoring the smeared images in the case when the smearing is rectilinear, horizontal, but nonuniform, in particular, piecewise uniform (example: cars on the road). It is proposed to solve a set of one-dimensional integral equations (IEs) (the first approach) or one two-dimensional IE (the second approach). In both approaches, the equations are not convolution type IEs; therefore, they are solved by the quadrature/cubature method with Tikhonov's regularization. The first approach is shown to be more preferable than the second one. Numerical examples confirm this. We have developed programs in the MATLAB system and we perform illustrative calculations.

In order to determine the image smear of a variable quantity $\Delta = \Delta(x)$, we propose a new modification of the "spectral method" (the "approach of divided spectra"), which enhances the image restoration accuracy.

It is also shown that to improve the image restoration quality, the image edge diffusing should be used for suppressing the Gibbs edge effect (the effect of false waves). The Gibbs effect may also be inner. In this case, it is suppressed by image smearing, which reduces the jumps in intensities.

We propose a new filter for eliminating impulse noise in the images—the modified median filter that suppresses noise, but almost does not distort the image.

In further publications, the proposed technique will be tested on the real images and compared with other methods.

The technique can be used in practice for restoring grouped images of several objects (e.g., people, planes, cars) moving with different speeds and therefore receiving different smears $\Delta$ in the images during exposure by a fixed camera.

**Author Contributions:** Conceptualization, V.S.; methodology, V.S. and A.D.; software, V.S.; formal analysis, V.S. and A.L.; data curation, V.S. and A.D.; writing—original draft preparation, V.S. and A.D.; writing—review and editing, V.S., A.D. and A.L.; visualization, V.S. and A.D. All authors have read and agreed to the published version of the manuscript.

**Funding:** This research was funded by MFCTC ITMO (Project Number: 619296).

**Conflicts of Interest:** The authors declare no conflict of interest.

## Appendix A

Let us briefly consider the question about terminology and image restoration methods. Many authors ([4,10–13,17,27], and others) use the terms blur, blurring, deblurring both for blurring and for defocusing of images. This creates an inconvenience. According to these terms, it is difficult to determine what kind of distortion (blurring or defocusing, etc.) is meant. In addition, the restoration of both blurry and defocused images is described only by two-dimensional integral equations (IEs). However, in this article, we show that in the case of nonuniform blur, it is better to use a set of one-dimensional IEs. We suppose that along with the terms blur, blurring, the terms such as smear, smearing and defocusing must be used, as well as one should solve both two-dimensional and one-dimensional IEs. It is also appropriate to use the complex term: smear (motion blur).

As stated in English Wikipedia [28], smear may refer to a sample smearing over a microscope slide, motion (in optics) that degrades sharpness of image, etc. In [29], it was also noted: in optics, image smear occurs when an object moves within the exposure time. In addition, in astronomy [30,31], time smearing or time-average smearing is the degradation of the reconstructed image of a celestial body observed by a ground-based interferometer that occurs because of the duration of the observation and rotation of the Earth.

Based on the foregoing, we can conclude that the terms smear and smearing can be used and is used quite often, along with the terms blur and blurring, in order to mean the image distortion during the mutual movement of the photographed object and the shooting equipment.

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
