# Peer review of "Eliminating Nonuniform Smearing and Suppressing the Gibbs Effect on Reconstructed Images†"

_computers, doi:10.3390/computers9020030_

Round 1

Reviewer 1 Report

The authors present methods to restore the smeared images. The topic is interesting and might be useful. A few suggestions which may further improve the manuscript.

  1. It might be better to change the tables of algorithms 1-5 to flowcharts, which may be easy to read and understand.
  2. The literature review in the introduction part is not very sufficient. The authors cited quite a few of their own work. They should include more work which can represent state-of-the-art techniques.
  3. The format of the manuscript has been off at multiple of places. For example, the paragraph alignment and line spaces are not consistent. Please be carefully proofread everything before submission.
  4. In Figure 1 and 3, it is hard to notice the difference between results (a) and (b). The authors may consider highlighting or annotating the different on the plots.

Author Response

  1. We agree that one ought to replace the tables of algorithms 1–5 to flowcharts. However, many authors use tables (structures, pseudo codes) of algorithms in their publications.

Examples: 1) Dell'Acqua P., Donatelli M., et al. Structure preserving preconditioners for image deblurring. J. Sci. Comput. 2017, 72(1), 147–171. doi: 10.1007/s10915-016-0350-2. 2) Article of Fergus R. [11]. We also decided to use the structures (pseudo codes) of the algorithms. In future publications, we will use the flowcharts for comparison.

  1. We agree that the literature review in the introduction is insufficient. We cited quite a few of own works. But one should include more works which can represent state-of-the-art techniques. Therefore, we have expanded the list of references.
  2. We cannot agree that the format of the manuscript has been off, the paragraph alignment and line spaces are not consistent. The manuscript was typed strictly according to the template (by file computer-template.dot). We sent to the editor a docx-file of the manuscript with aligned lines and paragraphs, but the editors inserted the line numbers and the format of the manuscript was violated.
  3. We agree that in Figure 1, it is hard to notice the difference between (a) and (b). The difference is only in the values ​​of the minimum and maximum intensities. However, this small difference is sufficient to make the image spectrum and the reconstructed image noticeably different (cf. Figures 10d and 2b, as well as Figures 10e and 2c). Very thin effect! As for Figure 3, then in it just the big differences between (a) and (b) are visible (the Gibbs effect in Figure 3b).

Sizikov V.S., Dovgan A.N., Lavrov A.V.

Thanks to reviewer 1 for helpful comments and suggestions.

Reviewer 2 Report

In this paper, the authors propose a method for restoring non-uniform motion blur. PSF-based inverse problem solving is a traditional approach, but solving a non-uniform problem is acceptable as a contribution point.

However, the performance evaluation of the study results has been qualitatively made, and it has not been presented with quantitative values such as SNR or PSNR. In addition, since the motion blurred images used in the experiment were generated, it is questionable how they can be applied in real cases. As a result, it is judged that the effectiveness verification for the proposed method is weak overall.

"Smearing" is a term that refers to a phenomenon that occurs when bright light is exposed in a low-light environment in a CCD sensor. Please consider using the term "motion blur".

Author Response

1) Reviewer 2 wrote that the performance evaluation of the study results has been qualitatively made, and it has not been presented with quantitative values such as SNR or PSNR. We do not agree. Indeed, in Section 4.4, formula (17) is given for the relative error σrel, and this is SNR and it is used in the manuscript.

2) Further, reviewer 2 wrote that the images were generated, it is questionable how they can be applied in real cases. We agree that the technique should be tested on real images. However, at first we develop a new technique, further we test it on semi-model images, and then we illustrate it on real images (in the next publication or report). No need to haste!

3) About the terms smearing, smear, blur, motion blur. Many (but not all) authors (Gonzalez, et al.) use only the terms blur and motion blur. As a result, both blurring and defocusing are described in the same terms, which creates inconvenience in perception. By these terms, it is difficult to determine what kind of distortion (blurring or defocusing) is meant. In addition, these distortions are described only by 2-dimensional integral equations. And in our article, it is shown that in the case of non-uniform smearing (motion blurring), it is better to use a set of 1-dimensional IEs. We will use the complex term: smear (motion blur). As follows from the English Wikipedia, the words smear and smearing are used to denote the image distortions during mutual movement of the photographed object and the shooting equipment.

Sizikov V.S., Dovgan A.N., Lavrov A.V.

Thanks to reviewer 2 for helpful comments and suggestions.

Round 2

Reviewer 1 Report

The authors modified a few places in the manuscript. The manuscript is improved a little compared with before.

Author Response

We thank Reviewer 1 for the second review.

Sizikov V.S., Dovgan A.N., Lavrov A.V.

Reviewer 2 Report

I agree that equation (17) has the same meaning as SNR.

I also think it is natural to apply it to semi-model images first. However, the actual motion blur may differ from the modeled image. It is not a haste, but a suggestion to apply it to some real sample images.

Since Wikipedia is not an official publication, it is difficult to present it as a reference for writing a thesis. I think smearing is a more ambiguous word than motion blur. However, this is minor. Respect the author's choice.

Author Response

We thank Reviewer 2 for the second review.

Sizikov V.S., Dovgan A.N., Lavrov A.V.
